# Evaluation and Comparison of Open and High-Resolution LULC Datasets for Urban Blue Space Mapping

Qi Zhou *[ID] and Xuanqiao Jing [ID]

School of Geography and Information Engineering, China University of Geosciences, Wuhan 430074, China
* Correspondence: zhouqi@cug.edu.cn

**Abstract:** Blue spaces (or water bodies) have a positive impact on the built-up environment and human health. Various open and high-resolution land-use/land-cover (LULC) datasets may be used for mapping blue space, but they have rarely been quantitatively evaluated and compared. Moreover, few studies have investigated whether existing 10-m-resolution LULC datasets can identify water bodies with widths as narrow as 10 m. To fill these gaps, this study evaluates and compares four LULC datasets (ESRI, ESA, FROM-GLC10, OSM) for blue space mapping in Great Britain. First, a buffer approach is proposed for the extraction of water bodies of different widths from a reference dataset. This approach is applied to each LULC dataset, and the results are compared in terms of accuracy, precision, recall, and the F1-score. We find that a high median accuracy (i.e., >98%) is achieved with all four LULC datasets. The OSM dataset gives the best recall and F1-score. Both the ESRI and ESA datasets produce better results than the FORM-GLC10 dataset. Additionally, the OSM dataset enables the identification of water bodies with widths of 10 m, whereas only water bodies with widths of 20 m or more can be identified in the other datasets. These findings may be beneficial for urban planners and designers in selecting an appropriate LULC dataset for blue space mapping.

**Keywords:** water body; land cover; land use; open data; OpenStreetMap

## 1. Introduction

The term "blue space" is used in the field of urban planning and design to refer to different kinds of water bodies (e.g., rivers, lakes, reservoirs, canals, open sea) in an urban built-up environment [1]. Extensive studies have reported that blue space has a positive impact in terms of reducing air pollution [2,3] and the urban heat island effect [4,5], as well as improving the physical and mental health of human beings [6–8]. The monitoring of urban blue space is also essential for achieving the 2030 Sustainable Development Goals (SDGs) adopted by the United Nations [9–11], i.e., SDG 6 (clear water and sanitation) and SDG 11 (sustainable cities and communities). It is therefore desirable to acquire suitable geospatial data for blue space mapping in order to support various applications.

Different data sources can be used for blue space mapping. Most existing studies used remote sensing (RS), i.e., the acquisition of information about an object (e.g., water body) without any physical contact with the object. For instance, Huang et al. [12] combined pixel- and object-based machine learning methods to identify water bodies from 2-m-resolution GeoEye-1 and WorldView-2 imagery, while Chen et al. [13] proposed a deep learning architecture for extracting urban water bodies from high-resolution (4–5 m) ZY-3 images. Chen et al. [14] recently proposed a method for detecting open water in urban areas based on high-resolution RS imagery. However, the use of RS requires a series of preprocessing steps (e.g., data acquisition, rectification, detection, and/or identification), which remains a technical challenge for most planners and designers. As an alternative, open land-use/land-cover (LULC) data products or geospatial data edited by global volunteers (e.g., OpenStreetMap, or OSM) have become essential sources for acquiring data related to water bodies. For instance, Feranec et al. [15] determined the changes and flows in European

landscapes between 1990 and 2000 based on the CORINE LULC data produced by the Land Monitoring Service. Teixeira et al. [16] used the CORINE LULC data to identify the forces driving changes in land cover, and Nowosad et al. [17] used the CCI-LC dataset, produced by the European Space Agency Climate Change Initiative, to quantitatively assess changes in land use (including crop, forest, grass, urban, water, and wetland) at a global scale. Recently, Long et al. [18] and Zhou et al. [19] have used the Finer Resolution Observation and Monitoring of Global Land Cover (FROM-GLC) dataset to investigate the spatial pattern of urban green space (all kinds of vegetations in an urban area, e.g., forest, grass, and shrub) at a global scale. Jakovljević et al. [20] used water bodies extracted from OSM data as a reference for comparison with water bodies extracted from different RS images (Sentinel-2, Landsat 8, and RapidEye). Luo et al. [21] used both the Joint Research Centre's Global Surface Water (JRC GSW) dataset and OSM data to study long-term (2008–2018) changes in the Yangtze River basin.

There are several benefits of using open LULC datasets for extracting urban water bodies. First, these datasets are generally produced by either non-profit organizations (e.g., Land Monitoring Service and European Space Agency) or global volunteers and are thus freely acquirable. Second, most open LULC datasets (e.g., CCI-LC and OSM) have global coverage, making it possible to acquire data related to urban blue space or water bodies at both regional and global scales. Third and most important, it is easy for planners and designers to extract water bodies from open LULC datasets, i.e., selecting one or several LULC types as water bodies. Compared with RS techniques, there are few technical challenges in using open LULC datasets. Despite the above-mentioned advantages, there are several limitations to open LULC data. First, an increasing number of open LULC datasets are becoming available. Although some studies have focused on comparing different LULC datasets [22,23], to the best of our knowledge, few studies have investigated how well these open LULC datasets perform in terms of blue space mapping and which dataset offers the optimal performance. Additionally, several 10-m-resolution (currently the highest spatial resolution) open LULC datasets (e.g., Esri 2020 Land Cover, ESA WorldCover, and FROM-GLC10) have recently been published for public use. Theoretically, it is possible to identify small water bodies as narrow as 10 m, but few studies have investigated whether a 10-m-resolution LULC dataset can identify water bodies of this size. The answers to the above-mentioned questions may be beneficial for planners and designers in selecting appropriate LULC datasets for blue space mapping.

To fill the above-mentioned research gaps, this study has two main objectives.

1.  Evaluate and compare a total of four global open LULC datasets (Esri 2020 Land Cover, ESA WorldCover, FROM-GLC10, and OSM) for urban blue space mapping, and determine which datasets give the best/worst performance.
2.  Investigate whether a 10-m-resolution LULC dataset can identify water bodies with a width of 10 m. If not, we determine the minimum width of water bodies that can be identified. This is achieved by proposing a simple approach for identifying water bodies of different widths.

## 2. Study Area and Data

### 2.1. Study Area

A total of 133 urban regions in Great Britain were chosen as the study areas (Figure 1). These urban areas were freely acquired from the urban center vector dataset of the GHS Settlement Model [24], which was developed by the European Commission's JRC. Such a large number of urban areas, rather than only a few, were chosen to reduce bias in the analysis. More importantly, a reference dataset related to water bodies is freely available for these urban areas, making it possible to compare among the water bodies extracted from different global open LULC datasets.

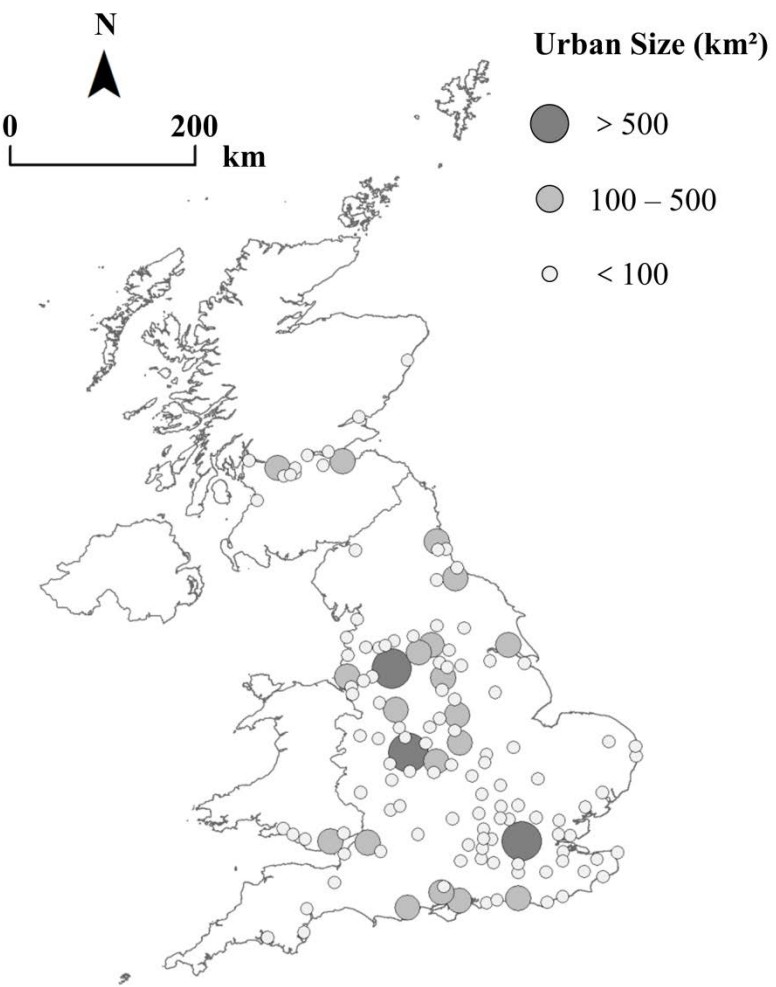

**Figure 1.** Locations of the 133 urban areas in Great Britain.

*2.2. Data*

(1) Global open LULC datasets

Four global open LULC datasets were selected for analysis (Table 1). These datasets were chosen because they each have a relatively high spatial resolution (10 m) or positional accuracy (e.g., <10 m). Specifically, the three raster datasets (ESRI, ESA, FROM-GLC10) were all produced using 10-m-resolution RS data (Sentinel-2) as the source. Although the OSM data may suffer from variable quality because they were edited by volunteers, previous studies have reported that the position accuracy of OSM data (in vector format) is mostly within 10 m [25–27]. Additionally, these four datasets were produced recently (from 2017–2021), raising the possibility of analyzing the performance of each dataset through comparisons with RS images in Google Earth. Finally, these datasets not only have global coverage but also are freely acquirable. The details of these open LULC datasets are introduced next:

- Esri 2020 Land Cover: This is a global 10-m-resolution LC dataset produced by Esri and published in June 2021 [28]. This dataset was first made available for the year 2020 and subsequently updated for five years from 2017–2021. In our study, the year-2020 dataset was used for analysis. Moreover, this dataset is divided into nine different LC types: *water, trees, flooded vegetation, crops, built area, bare ground, snow/ice, clouds,* and *rangeland*. The LC type *water* was extracted and assumed to represent water bodies.
- ESA WorldCover: This is another global 10-m-resolution LC dataset, produced by the European Space Agency and published in October 2021 [29]. This dataset was made available for 2020 and includes 11 different LC types: *tree cover, shrubland, grassland, cropland, built-up, bare/sparse vegetation, snow and ice, permanent water bodies, herbaceous*

*wetland, mangroves,* and *moss and lichen.* The LC type *permanent water bodies* was extracted for analysis.

- FROM-GLC10: This global 10-m-resolution LC dataset was produced by Tsinghua University and published in March 2019 [30]. This dataset was made available for 2017 and includes 10 different LC types: *cropland, forest, grassland, shrubland, wetland, water, tundra, impervious surface, bareland,* and *snow/ice*. The LC type *water* was extracted for subsequent analysis.
- OSM: This global open dataset is represented in vector format. The OSM data of different map features or layers (e.g., *roads, buildings, landuse, natural,* and *water*) can be acquired from a third-party platform, Geofabrik. Moreover, this platform provides datasets for different countries and regions across the globe. For this study, the five different components of the *water* layer (*dock, reservoir, river, riverbank, water*) were acquired in December 2020 and assumed to represent water bodies.

**Table 1.** Description of the four global open LULC datasets.

| Name | Format | Spatial Resolution | Year | LULC Types | Website |
|---|---|---|---|---|---|
| Esri 2020 Land Cover (ESRI) | Raster | 10 m | 2020 | *Water, trees, flooded vegetation, crops, built area, bare ground, snow/ice, clouds, rangeland* | https://livingatlas.arcgis.com/landcover/ accessed on 20 February 2022 |
| ESA WorldCover (ESA) | Raster | 10 m | 2020 | *Tree cover, shrubland, grassland, cropland, built-up, bare/sparse vegetation, snow and ice, permanent water bodies, herbaceous wetland, mangroves, moss and lichen* | https://esa-worldcover.org/en accessed on 25 January 2022 |
| Finer Resolution Observation and Monitoring-Global Land Cover (FROM-GLC10) | Raster | 10 m | 2017 | *Cropland, forest, grassland, shrubland, wetland, water, tundra, impervious surface, bareland, snow/ice* | http://data.ess.tsinghua.edu.cn accessed on 30 January 2022 |
| OpenStreetMap (OSM) | Vector | N/A | 2020 | *Dock, reservoir, river, riverbank, water* | https://download.geofabrik.de accessed on 30 December 2020 |

(2) Reference data

Ordnance Survey (OS) data, produced by the national mapping agency of Great Britain, was used as the reference. These data were acquired in vector format at a 1:10,000 scale; more importantly, they are "the most detailed 'street level' mapping product available within the open data arena" [31]. The OS data include 20 different LU types. The *SurfaceWater_Area* and *TidalWater* (*TidalWater* denotes the extent of tidal water up to the high water mark and normal tide limit of rivers [31]) regions were extracted and assumed to represent water bodies.

## 3. Methods

This study has two objectives. The first is to compare the water bodies extracted from different LULC datasets and to investigate which LULC dataset gives the best performance. The second objective is to investigate whether the existing 10-m-resolution LULC datasets can detect water bodies with a width of 10 m. If not, we wish to determine the minimum width at which water bodies can be detected. To answer these two research questions, water bodies of various widths were extracted from the reference dataset, and these were compared with those extracted from each global open LULC dataset in terms of various measures.

### 3.1. Extracting Water Bodies of Different Widths

We propose a buffer-based approach for extracting water bodies (of different widths) from the reference dataset. To illustrate this approach, a schematic figure is presented in Figure 2. This figure shows two water bodies, *A* and *B* (Figure 2a).

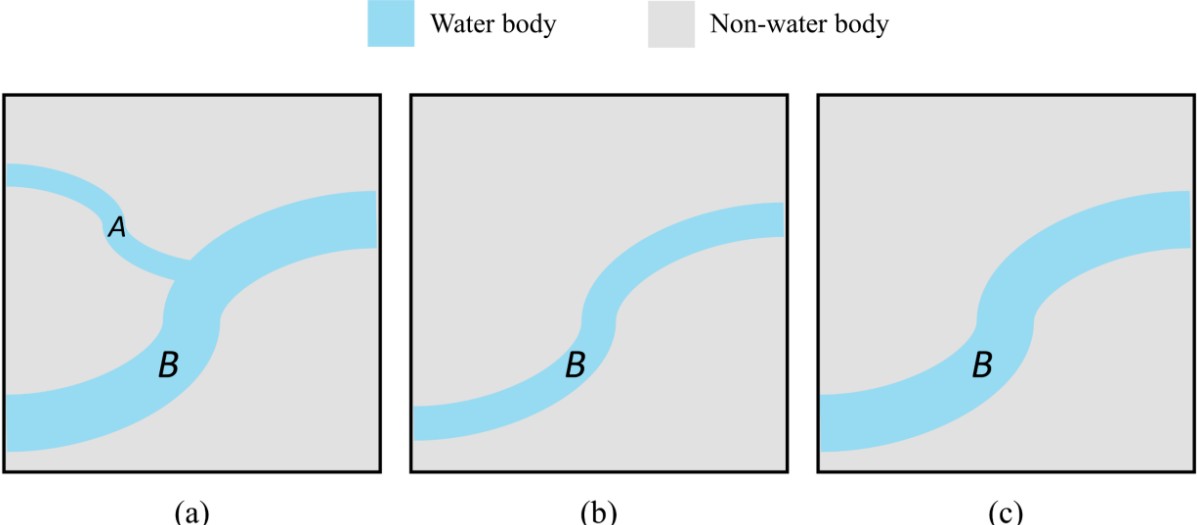

**Figure 2.** Illustration of the principle of extracting water bodies: (**a**) original; (**b**) after the 1st buffering; (**c**) after the 2nd buffering.

The proposed approach has two main steps.

First, we buffer these water bodies with a negative buffer radius (e.g., −10 m). Thus, water body *A*, which has a smaller width, is removed, and only water body *B* is retained (Figure 2b). We then buffer the remaining water bodies with a positive buffer radius (e.g., 10 m). Thus, retained water body *B* is restored to its original size.

Different thresholds can be used for the buffer radius (e.g., 0 m, 5 m, 10 m, 15 m, 20 m, and 25 m). The threshold approximately equals half of the water body width. Thus, from the same reference dataset, sub-datasets with different water body widths can be extracted. For example, if the threshold is set to 10 m, the corresponding sub-dataset contains water bodies with a width of 20 m or more.

### 3.2. Evaluating Various LULC Datasets with Different Measures

Four common measures are used to quantitatively compare the water bodies extracted from the global open LULC datasets with those extracted from the reference dataset. These measures are *accuracy*, *precision*, *recall*, and the F1-score, which are widely used to evaluate the effectiveness of classification models [32,33]. These metrics are defined as follows:

$$Accuracy = \frac{TP + TN}{TP + FN + TN + FP} \times 100\% \tag{1}$$

$$Precision = \frac{TP}{FP + TP} \times 100\% \tag{2}$$

$$Recall = \frac{TP}{FN + TP} \times 100\% \tag{3}$$

$$\text{F1-score} = \frac{2 \times Precision \times Recall}{Precision + Recall} \tag{4}$$

where *TP* (true positive) denotes the common area between water bodies extracted from an open LULC dataset and the reference dataset, *TN* (true negative) denotes the common area between non-water bodies (regions not classified as water bodies) in both datasets, *FP* (false positive) denotes the total area of water bodies extracted from the open dataset

but not from the reference dataset, and *FN* (false negative) denotes the total area of water bodies extracted from the reference dataset but not from the open dataset.

## 4. Results and Analysis

First, because most of the LC datasets have a spatial resolution of 10 m, a sub-dataset was extracted from the reference dataset using a 5-m buffer radius. This sub-dataset was then compared with each LULC dataset in terms of *accuracy*, *precision*, *recall*, and the F1-score. The evaluation results for the 133 urban areas are presented in Figure 3.

From Figure 3, the following conclusions can be stated:

1.  The accuracy is high (e.g., >90%) for most of the 133 urban areas, although this is not always the case for urban areas along the coastline. This is probably because, in urban areas, most land is correctly classified as non-water bodies.
2.  The precision is generally high (e.g., >60%) in most urban areas. This indicates that most water bodies extracted from the various global open LULC datasets are also identified as water bodies in the corresponding reference dataset.
3.  The recall is relatively low (e.g., <60%) for some urban areas. This indicates that several water bodies in the reference dataset were not correctly identified as water bodies in the LULC datasets. Moreover, in terms of the three LC datasets (ESRI, ESA, and FROM-GLC10), the urban areas with a relatively low recall are mostly located in central regions of Great Britain. In terms of the OSM dataset, areas with a relatively low recall are mostly located along the boundary (i.e., coastline) of Great Britain. This indicates that the weaknesses of using different LULC datasets for blue space mapping may vary (Figure 4). Specifically, three LC datasets (ESRI, ESA, and FROM-GLC10) cannot identify some water bodies with relatively small widths (e.g., 10–20 m, Figure 4a–c). Although this is not the case for the OSM dataset (Figure 4d), water bodies in the open sea cannot be identified in the OSM data (Figure 4i) but can be identified by the other three global open LC datasets (Figure 4f–h).
4.  The F1-score is relatively low for some urban areas. The spatial pattern of the F1-score is similar to that of the recall, which indicates that the F1-score is highly dependent on the recall because the precision is relatively high.

Figure 5 compares the water bodies extracted from the four global open LULC datasets with those extracted from the reference dataset at six different buffer thresholds (from 0 to 25 m at intervals of 5 m). These thresholds correspond to water body widths varying from 0 to 50 m at intervals of 10 m. The corresponding values of the various metrics are listed in Appendix A.

Figure 5 and Appendix A indicate the following:

1.  In terms of accuracy, the median values for the various LULC datasets are high, i.e., 98% or above. This indicates that most of the land in urban areas can be correctly classified as either water bodies or non-water bodies. Nevertheless, the minimum value is much lower (i.e., less than 70%) using the OSM dataset than with the other three LC datasets. This is because the open sea adjacent to some urban areas cannot be identified in the OSM data, as shown in Figure 4.
2.  In terms of precision, the median value varies under different buffer thresholds and with different global open LULC datasets. As an example, when using the OSM dataset, the median value is higher than 92% with a buffer threshold of 0 m, but this value decreases to 56% with a buffer threshold of 25 m. This indicates that water bodies with a width of 10 m or less can be identified using the OSM dataset. In contrast, when using the FROM-GLC10 dataset, all median values are greater than 98%, regardless of the buffer threshold. This is probably because, with this dataset, few water bodies with a width of 50 m or less are identified. Moreover, the median value varies with different LULC datasets. Generally, the greatest median value comes from using the FROM-GLC10 dataset (99%) or the ESA dataset (95%) rather than the OSM dataset (92%) or the ESRI dataset (84%). This indicates that the FROM-GLC10 dataset performs the best and the ESRI dataset performs the worst in terms of precision.

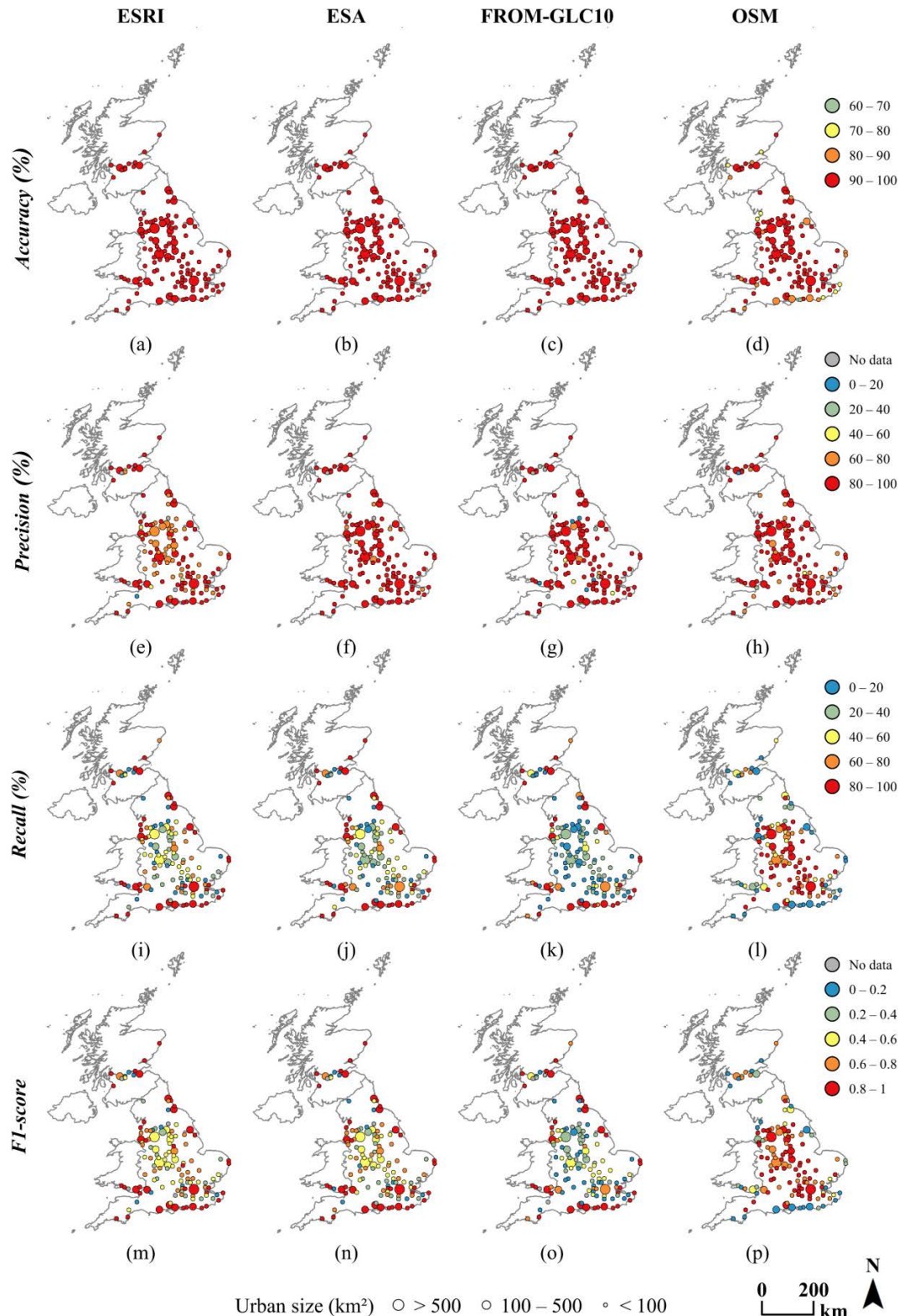

**Figure 3.** Comparisons of four global open LULC datasets (ESRI, ESA, FROM-GLC10, and OSM) with the reference sub-dataset extracted using a 5-m buffer radius: (**a**) ESRI, accuracy; (**b**) ESA, accuracy; (**c**) FROM-GLC10, accuracy; (**d**) OSM, accuracy; (**e**) ESRI, precision; (**f**) ESA, precision; (**g**) FROM-GLC10, precision; (**h**) OSM, precision; (**i**) ESRI, recall; (**j**) ESA, recall; (**k**) FROM-GLC10, recall; (**l**) OSM, recall; (**m**) ESRI, F1-score; (**n**) ESA, F1-score; (**o**) FROM-GLC10, F1-score; (**p**) OSM, F1-score. "No data" means that no water bodies were identified in that urban area using the corresponding LULC dataset.

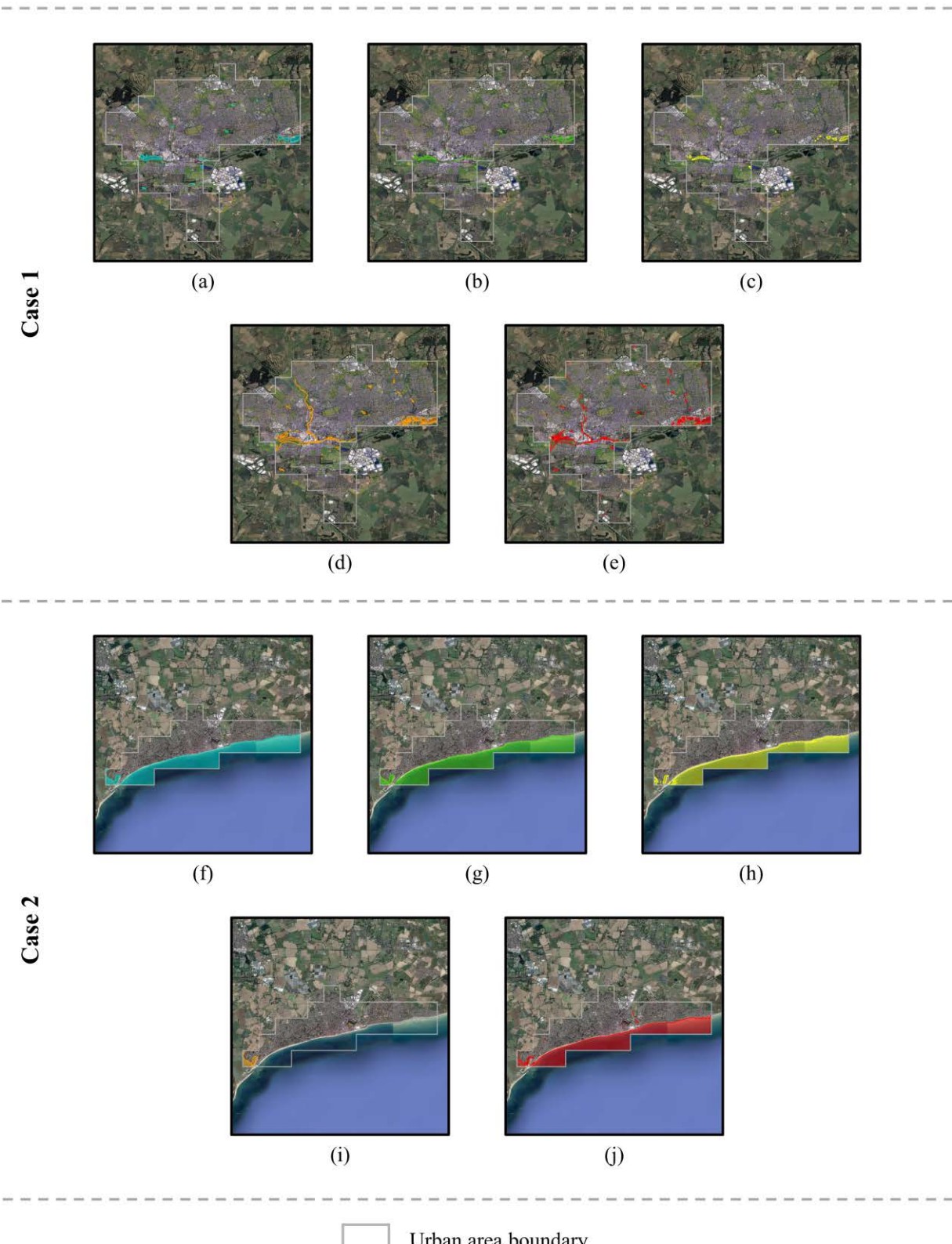

**Figure 4.** Illustration of the weaknesses of using global open LULC datasets for blue space mapping: (**a**) Northampton, ESRI; (**b**) Northampton, ESA; (**c**) Northampton, FROM-GLC10; (**d**) Northampton, OSM; (**e**) Northampton, reference; (**f**) Bognor Regis, ESRI; (**g**) Bognor Regis, ESA; (**h**) Bognor Regis, FROM-GLC10; (**i**) Bognor Regis, OSM; (**j**) Bognor Regis, reference.

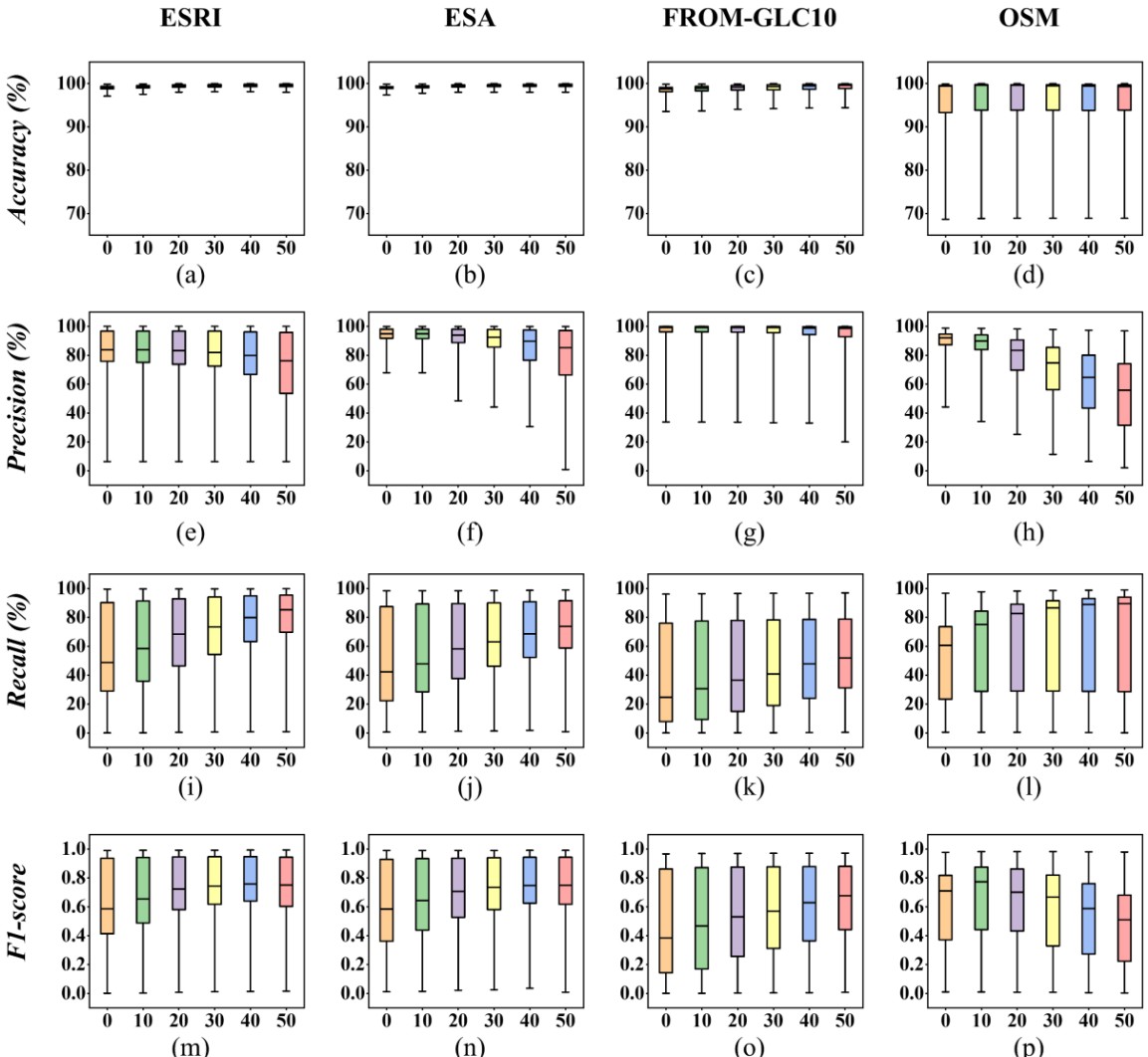

**Figure 5.** Comparison of the four global open LULC datasets (ESRI, ESA, FROM-GLC10, and OSM) with reference sub-datasets extracted using different buffer thresholds, visualized using box plots: (**a**) ESRI, accuracy; (**b**) ESA, accuracy; (**c**) FROM-GLC10, accuracy; (**d**) OSM, accuracy; (**e**) ESRI, precision; (**f**) ESA, precision; (**g**) FROM-GLC10, precision; (**h**) OSM, precision; (**i**) ESRI, recall; (**j**) ESA, recall; (**k**) FROM-GLC10, recall; (**l**) OSM, recall; (**m**) ESRI, F1-score; (**n**) ESA, F1-score; (**o**) FROM-GLC10, F1-score; (**p**) OSM, F1-score. The *x*-axis denotes the minimum width of the water bodies, i.e., varying from 0 to 50 m at intervals of 10 m.

3.  In terms of recall, the median value generally increases with increasing buffer threshold for the various LULC datasets. For instance, using the ESRI dataset, the value is close to 50% when the buffer threshold is 0 m, but this value increases to 80% or more when the buffer threshold reaches 20 or 25 m. This indicates that the ESRI dataset may fail to detect some water bodies with a relatively small width (e.g., 0–20 m). A similar conclusion can be reached for the other three LUCL datasets. Nevertheless, the greatest median value is much higher when using the OSM dataset (90%) or the ESRI dataset (85%) compared with the ESA dataset (74%) or the FROM-GLC10 dataset (52%). Thus, the OSM dataset gives the best performance and the FROM-GLC10 exhibits the worst performance in terms of recall.

4.  In terms of the F1-score, the highest median value of 0.77 occurs when using the OSM dataset; median values of 0.68–0.76 are given by the other three LC datasets. Moreover, using the OSM dataset, the maximum F1-score is achieved when the buffer threshold is set to 5 m. This indicates that the OSM dataset can detect water bodies

with a width of around 10 m. In contrast, the other three LC datasets attain maximum F1-scores with a buffer threshold of 20–25 m. This suggests that these datasets can only effectively identify water bodies with widths of 40–50 m.

Figure 6 shows some typical examples of using the four global open LULC datasets for blue space mapping. Specifically, each dataset was overlapped with RS images in Google Earth, and water bodies of different widths (0–50 m) were involved in the analysis.

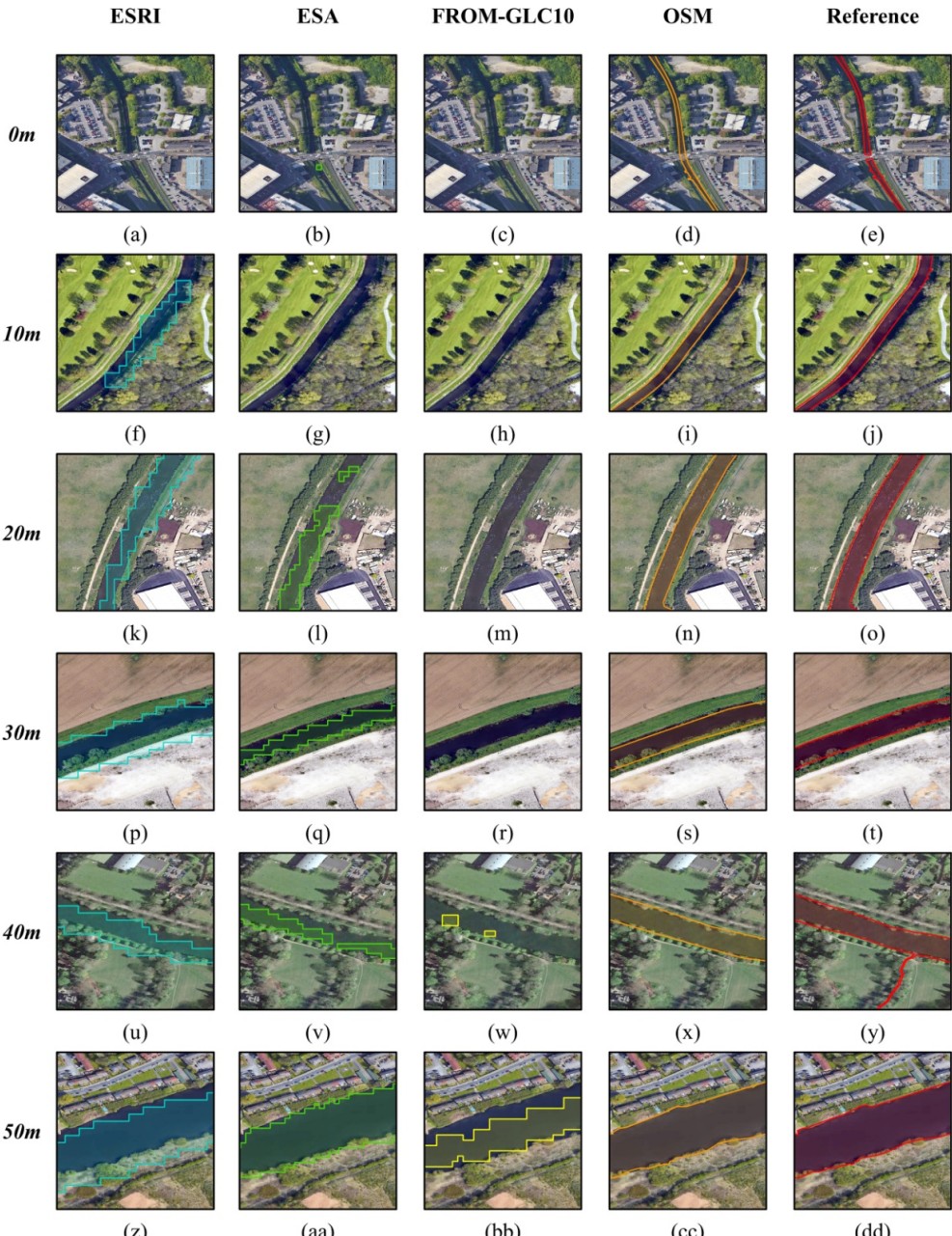

**Figure 6.** Comparison of using the four global open LULC datasets (ESA, ESRI, FROM-GLC, and OSM, highlighted in different colors) for blue space mapping by overlapping with Google Earth images. Different width thresholds were used for the analysis: (**a**–**e**) 0 m; (**f**–**j**) 10 m; (**k**–**o**) 20 m; (**p**–**t**) 30 m; (**u**–**y**) 40 m; (**z**–**dd**) 50 m.

We can see from Figure 6 that water bodies of less than 10 m width cannot be identified using the ESRI, ESA, and FROM-GLC10 datasets (Figure 6a–c). This is because the spatial resolution of these datasets is 10 m. Moreover, water bodies with widths of 10–40 m may not be effectively extracted (Figure 6f,l,w) or even identified (Figure 6g,m,r). Thus, we

conclude that a 10-m-resolution LC dataset may not enable the identification of water bodies with a width of 10 m. Specifically, a 10-m-resolution LC dataset can identify water bodies with larger widths, i.e., 20–40 m, depending on the dataset. For instance, both the ESRI and ESA datasets allow water bodies with a width of 20 m to be identified, but this threshold increases to 40 or 50 m when using the FROM-GLC10 dataset.

However, this is not the case for the OSM dataset. That is, with this dataset, water bodies with widths of less than 10 m can be identified (Figure 6e,j). This is probably because most of the OSM data were provided and/or edited through data vectorization based on Bing Image with a spatial resolution as high as 0.6 m [34]. Thus, the use of the OSM dataset provides the best blue space mapping, which is consistent with the results in Figure 5m–p.

When using the ESRI dataset, some non-water bodies may also be identified (Figure 6p,u,z). Therefore, this dataset achieves a relatively low precision (Figure 5).

## 5. Discussion

### 5.1. Implications

This study compared four global open and high-resolution LULC datasets (ESRI, ESA, FROM-GLC10, and OSM) for blue space mapping in terms of their accuracy, precision, recall, and F1-score. Although other studies have compared different open LCLU datasets [35–37], most of these studies involved lower-resolution datasets (e.g., 20–100 m). This study found that all of the analyzed datasets achieve excellent accuracy. The OSM dataset performed the best in terms of recall and the F1-score, while the FROM-GLC10 performed the worst in terms of recall and the F1-score. However, FROM-GLC10 performed the best in terms of precision because fewer water bodies were identified using this dataset. These findings are not fully consistent with those of existing studies. For instance, Liao et al. [33] reported that the FROM-GLC10 dataset gives the best performance for urban green space mapping in terms of accuracy, recall, and the F1-score. This indicates that the effectiveness of using an LULC dataset may vary according to the application (e.g., blue space or green space mapping).

Furthermore, we investigated whether a 10-m-resolution LC dataset can be used to identify water bodies with a width of 10 m. We found that few 10-m-resolution LC datasets can accurately identify water bodies with widths of between 0 and 20 m, and the specific width that can be identified varies among the different LULC datasets. Specifically, the OSM dataset was able to identify water bodies with a width of 10 m or less, and both the ESRI and ESA datasets identified water bodies with widths of around 20–30 m, but the FROM-GLC10 dataset could only identify water bodies 40–50 m in width (Figure 6). Thus, both ESRI and ESA performed better than FROM-GLC10 in terms of identifying small water bodies. Moreover, Table 2 lists the area percentages of water bodies of the two urban areas shown in Figure 4 (Northampton and Bognor Regis), which were calculated using both the open LULC datasets and the reference dataset. This table indicates that, for the study area of Northampton, the use of the OSM dataset gives the best performance for blue space mapping, because the area percentage of water bodies (1.52%) calculated using this dataset is closest to that (1.51%) of the corresponding reference dataset. For Bognor Regis, however, the use of the OSM dataset produces the worst performance, because the open sea is not identified using this dataset (Figure 4). This indicates that, in practical applications, it may be better to integrate the OSM dataset with another LC dataset(s) for blue space mapping.

To the best of our knowledge, this is the first time that the performance of various LULC datasets has been compared for the purpose of blue space mapping. The results may be beneficial for urban planners and designers in selecting appropriate datasets (e.g., OSM and/or ESRI) for blue space mapping. Indeed, a recent study reported that the ESRI dataset can be used to acquire urban blue space for investigating land surface temperature [38]. Additionally, the ESRI data product includes data for each year from 2017–2021. Thus, this dataset may be used to analyze changes in land use (e.g., ecosystem accounting [39,40]). In contrast, both the ESA and FROM-GLC10 datasets are only available for a single year.

**Table 2.** Area percentage of water bodies of the two urban areas in Figure 4, calculating using different LULC datasets.

| Study Area | Measure | Open LULC Dataset | | | | Reference Dataset |
|---|---|---|---|---|---|---|
| | | ESRI | ESA | FROM-GLC10 | OSM | |
| Case 1 Northampton | Area percentage of water bodies | 0.84% | 0.59% | 0.42% | 1.52% | 1.51% |
| Case 2 Bognor Regis | | 31.92% | 31.60% | 29.67% | 1.14% | 31.85% |

*5.2. Limitations*

There are several limitations of this study. First, there are numerous LULC datasets that may be used for blue space mapping. For instance, CCI-LC is a 300-m-resolution LC dataset produced by the ESA, and GlobeLand30 is a 30-m-resolution LC dataset produced by the National Geomatics Center of China. These datasets were not analyzed because they have relatively low spatial resolutions (i.e., 30–300 m) and thus they would theoretically perform worse than the datasets analyzed in this study. Indeed, Mao et al. [41] found that existing 30-m-resolution global water body datasets experience difficulties when mapping small rivers (i.e., widths narrower than 300 m). Nevertheless, other datasets should be incorporated into the analysis in future work.

Second, there are a number of LULC types in each LULC dataset. We mainly extracted areas of *water* from the LC datasets and used the five different types of water bodies (*dock, reservoir, river, riverbank,* and *water*) in the OSM dataset. These types are conceptually defined as permanent water areas, rather than seasonal water areas (e.g., wetland), and were visually determined through comparisons with Google Earth images (Figure 7). However, the evaluation results may vary if different LULC types were selected for the analysis [33]. Therefore, in future work, it would be interesting to compare the performance when different LULC types are extracted as water bodies for blue space mapping.

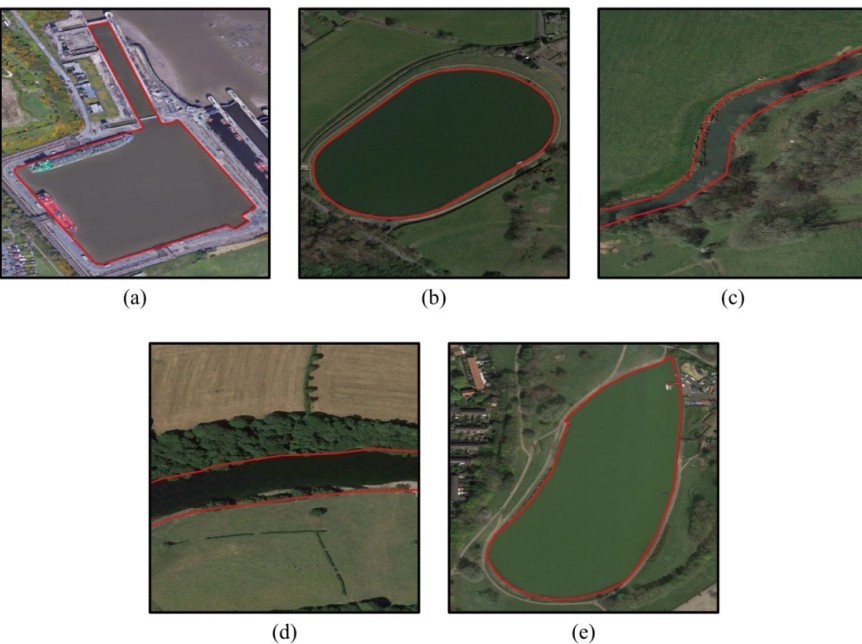

**Figure 7.** Overlapping Google Earth images with OSM data (highlighted in red) of different types: (**a**) dock; (**b**) reservoir; (**c**) river; (**d**) riverbank; (**e**) water.

Finally, this study only considered 133 urban areas of Great Britain for the analysis. The water bodies in rural areas were not analyzed because urban areas have much higher

populations. Existing studies have highlighted the positive impact of water bodies (or blue space) on the physical and mental health and well-being of urban residents [6–8]. It is therefore more important to investigate the distribution of water bodies in urban built-up areas. It remains necessary to validate our results with urban areas in other countries and regions. This is especially true for the OSM dataset because existing studies have reported that both the map scale and the completeness (how well a region has been mapped) of OSM data may vary dramatically in different countries and regions [42–47]. Specifically, Zhou et al. [46] reported that the completeness of OSM LULC data is higher in Europe than in other regions. This indicates that the findings when using OSM data may not be applicable to other study areas. Recall that this study involved only Great Britain as the study area due to the availability of the reference dataset. However, water bodies in rural areas and in other countries and regions should be considered for further validation.

## 6. Conclusions

This study quantitatively compared four global open and high-resolution LULC datasets (ESRI, ESA, FROM-GLC10, and OSM) for urban blue space mapping. These datasets were evaluated by comparing their results with those from a reference dataset acquired from the national mapping agency of Great Britain (Ordnance Survey) across four different metrics (accuracy, precision, recall, and F1-score). Specifically, 133 urban regions were chosen as study areas, and sub-datasets were extracted from the reference dataset to represent water bodies of various widths (varying from 0 to 50 m at intervals of 10 m). The results led to the following conclusions:

1. All water bodies extracted from these LULC datasets achieved good performance in terms of accuracy. The OSM dataset performed the best in terms of recall and the F1-score. The FORM-GLC10 dataset performed the worst in terms of recall and the F1-score, although it offered the best performance in terms of precision.
2. The OSM dataset identified water bodies with a width of 10 m or less. The ESRI and ESA datasets could only identify water bodies with widths of more than 10 m (e.g., 20–30 m). The FROM-GLC10 dataset was only able to identify water bodies with a width of 40–50 m.

We conclude that the OSM dataset performs better than existing LC datasets for blue space mapping. Both the ESRI and ESA datasets outperform the FROM-GLC10 dataset.

In future work, other global or regional open LULC datasets will be included in the analysis. Different LULC types may also be selected as water bodies to investigate the performance of various LULC types for blue space mapping. Finally, the water bodies in other countries and regions will be analyzed to validate whether the conclusions from this study are more widely applicable.

**Author Contributions:** Conceptualization, methodology, writing—review and editing, Q.Z.; Data curation, formal analysis, visualization, writing—original draft preparation, X.J. All authors have read and agreed to the published version of the manuscript.

**Funding:** This research was funded by National Natural Science Foundation of China (No. 41771428).

**Institutional Review Board Statement:** Not applicable.

**Informed Consent Statement:** Not applicable.

**Acknowledgments:** We would like to express special thanks to the editor and all the anonymous reviewers for their valuable comments that have helped improve this paper substantially.

**Conflicts of Interest:** The authors declare no conflict of interest.

### Appendix A. Comparing the Four Global Open LULC Datasets (ESRI, ESA, FROM-GLC10, and OSM) with Reference Sub-Datasets Extracted Using Different Buffer Thresholds

| Data | Quartile | Accuracy (%) | | | | | | Precision (%) | | | | | | Recall (%) | | | | | | F1-Score | | | | | |
|---|---|---|---|---|---|---|---|---|---|---|---|---|---|---|---|---|---|---|---|---|---|---|---|---|---|
| | | 0 m | 5 m | 10 m | 15 m | 20 m | 25 m | 0 m | 5 m | 10 m | 15 m | 20 m | 25 m | 0 m | 5 m | 10 m | 15 m | 20 m | 25 m | 0 m | 5 m | 10 m | 15 m | 20 m | 25 m |
| ESRI | Min. | 97.11 | 97.52 | 98.01 | 98.10 | 98.13 | 97.96 | 6.45 | 6.45 | 6.45 | 6.45 | 6.45 | 6.45 | 0.10 | 0.15 | 0.41 | 0.62 | 0.79 | 0.91 | 0.00 | 0.00 | 0.01 | 0.01 | 0.01 | 0.02 |
| | Q1 | 98.79 | 98.98 | 99.19 | 99.23 | 99.27 | 99.27 | 75.76 | 75.08 | 73.80 | 72.56 | 66.82 | 53.71 | 29.19 | 35.98 | 46.55 | 54.54 | 63.58 | 69.80 | 0.42 | 0.49 | 0.58 | 0.62 | 0.64 | 0.60 |
| | Median | 99.08 | 99.26 | 99.47 | 99.53 | 99.58 | 99.61 | 83.82 | 83.72 | 83.29 | 81.95 | 79.80 | 76.15 | 48.75 | 58.41 | 68.51 | 73.36 | 79.97 | 85.20 | 0.59 | 0.65 | 0.72 | 0.74 | 0.76 | 0.75 |
| | Q3 | 99.32 | 99.52 | 99.69 | 99.77 | 99.82 | 99.83 | 96.70 | 96.70 | 96.68 | 96.62 | 96.08 | 95.78 | 90.21 | 91.20 | 92.89 | 94.05 | 94.88 | 95.33 | 0.93 | 0.94 | 0.94 | 0.95 | 0.95 | 0.94 |
| | Max. | 99.86 | 99.93 | 99.95 | 99.96 | 99.98 | 99.98 | 100.00 | 100.00 | 100.00 | 100.00 | 100.00 | 100.00 | 99.60 | 99.64 | 99.66 | 99.69 | 99.76 | 99.82 | 0.99 | 0.99 | 0.99 | 0.99 | 0.99 | 0.99 |
| ESA | Min. | 97.36 | 97.76 | 97.97 | 97.98 | 97.99 | 98.01 | 67.93 | 67.89 | 48.52 | 44.12 | 30.66 | 0.91 | 0.59 | 0.72 | 1.14 | 1.33 | 1.83 | 0.89 | 0.01 | 0.01 | 0.02 | 0.03 | 0.04 | 0.01 |
| | Q1 | 98.83 | 99.04 | 99.21 | 99.26 | 99.32 | 99.34 | 91.82 | 91.61 | 88.75 | 85.71 | 76.63 | 66.41 | 22.29 | 28.47 | 37.69 | 46.17 | 52.37 | 58.85 | 0.36 | 0.44 | 0.53 | 0.58 | 0.63 | 0.62 |
| | Median | 99.11 | 99.26 | 99.46 | 99.57 | 99.61 | 99.62 | 94.93 | 94.82 | 94.03 | 92.51 | 89.68 | 85.25 | 42.32 | 47.96 | 58.14 | 63.11 | 68.56 | 73.80 | 0.58 | 0.64 | 0.71 | 0.73 | 0.75 | 0.75 |
| | Q3 | 99.31 | 99.48 | 99.67 | 99.75 | 99.81 | 99.84 | 98.09 | 98.08 | 98.01 | 97.95 | 97.57 | 97.15 | 87.43 | 89.30 | 89.54 | 90.01 | 90.77 | 91.57 | 0.93 | 0.93 | 0.94 | 0.94 | 0.94 | 0.94 |
| | Max. | 99.85 | 99.93 | 99.94 | 99.95 | 99.96 | 99.97 | 99.92 | 99.92 | 99.92 | 99.91 | 99.90 | 99.88 | 98.43 | 98.47 | 98.49 | 98.52 | 98.69 | 98.99 | 0.99 | 0.99 | 0.99 | 0.99 | 0.99 | 0.99 |
| FROM-GLC10 | Min. | 93.54 | 93.66 | 94.04 | 94.22 | 94.34 | 94.44 | 33.76 | 33.74 | 33.71 | 33.31 | 33.08 | 20.03 | 0.05 | 0.07 | 0.17 | 0.19 | 0.27 | 0.41 | 0.00 | 0.00 | 0.00 | 0.00 | 0.01 | 0.01 |
| | Q1 | 98.14 | 98.34 | 98.51 | 98.57 | 98.68 | 98.87 | 96.22 | 96.19 | 96.10 | 95.76 | 94.86 | 93.34 | 7.86 | 9.41 | 15.51 | 19.70 | 24.59 | 32.00 | 0.14 | 0.17 | 0.26 | 0.32 | 0.37 | 0.45 |
| | Median | 98.71 | 98.92 | 99.16 | 99.31 | 99.46 | 99.60 | 99.15 | 99.13 | 99.12 | 99.05 | 98.80 | 98.54 | 24.73 | 30.69 | 36.63 | 40.84 | 47.86 | 51.96 | 0.38 | 0.47 | 0.53 | 0.57 | 0.63 | 0.68 |
| | Q3 | 99.10 | 99.35 | 99.58 | 99.69 | 99.76 | 99.82 | 99.84 | 99.84 | 99.83 | 99.79 | 99.74 | 99.65 | 73.98 | 75.53 | 75.85 | 76.19 | 76.77 | 77.45 | 0.85 | 0.86 | 0.86 | 0.86 | 0.87 | 0.87 |
| | Max. | 99.84 | 99.91 | 99.93 | 99.96 | 99.98 | 99.98 | 100.00 | 100.00 | 100.00 | 100.00 | 100.00 | 100.00 | 96.12 | 96.45 | 96.54 | 96.65 | 96.73 | 96.98 | 0.97 | 0.97 | 0.97 | 0.97 | 0.97 | 0.97 |
| OSM | Min. | 68.69 | 68.89 | 68.92 | 68.90 | 68.90 | 68.90 | 44.22 | 34.19 | 25.26 | 11.48 | 6.50 | 2.11 | 0.54 | 0.50 | 0.45 | 0.40 | 0.29 | 0.17 | 0.01 | 0.01 | 0.01 | 0.01 | 0.01 | 0.00 |
| | Q1 | 94.85 | 95.12 | 95.12 | 95.12 | 95.11 | 95.14 | 87.53 | 84.13 | 70.00 | 56.25 | 43.86 | 33.07 | 24.47 | 28.84 | 32.87 | 35.74 | 34.43 | 33.33 | 0.38 | 0.44 | 0.44 | 0.34 | 0.27 | 0.23 |
| | Median | 99.42 | 99.56 | 99.56 | 99.50 | 99.40 | 99.30 | 92.13 | 89.85 | 83.35 | 74.68 | 64.80 | 55.87 | 60.65 | 75.16 | 82.61 | 86.52 | 88.88 | 89.55 | 0.71 | 0.77 | 0.70 | 0.67 | 0.59 | 0.51 |
| | Q3 | 99.58 | 99.75 | 99.74 | 99.71 | 99.69 | 99.64 | 94.72 | 94.09 | 90.01 | 85.03 | 80.04 | 74.09 | 73.50 | 84.03 | 89.16 | 91.43 | 93.07 | 93.95 | 0.82 | 0.87 | 0.86 | 0.82 | 0.76 | 0.68 |
| | Max. | 99.87 | 99.97 | 99.94 | 99.94 | 99.93 | 99.94 | 98.71 | 98.63 | 98.28 | 97.82 | 97.33 | 96.85 | 96.73 | 97.71 | 98.22 | 98.57 | 98.81 | 99.03 | 0.98 | 0.98 | 0.98 | 0.98 | 0.98 | 0.98 |

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
