# Peer review of "Evaluation and Comparison of Open and High-Resolution LULC Datasets for Urban Blue Space Mapping"

_remotesensing, doi:10.3390/rs14225764_

Round 1

Reviewer 1 Report

The authors present an evaluation of four global land-use land-cover maps for the mapping of urban blue space. They find that OSM performs better than the satellite remote sensing based maps, and is particularly good at mapping small (<10m width) water bodies. I find the results and conclusions very relevant and important, however I find a serious flaw in the analysis which needs to be addressed before the paper can be published. I explain below.

The main concern I have is that the authors use different definitions of “blue space” when selecting the LCLU classes form the global datasets. This results in confounded comparisons and incorrect accuracy assessments. Firstly, it is not clear why the authors include flooded vegetation and wetland for the Esri, ESA WorldCover and FROM-GLC10 maps, but they exclude wetland in the OSM map. This is inconsistent and needs to be addressed. I would argue that the authors come up with a more rigorous definition of “blue space” to include permanent water bodies that exclude wetlands (which are typically ephemerally covered with water). Secondly, if the authors are going to include flooded vegetation/wetland in their definition, then they need to also include mangroves for the ESA WorldCover definition because this aligns with the flooded vegetation definition in Esri.

I am also concerned that, as it stands, the discussion section is weak. The authors do not place their findings in the context of the other literature. They do not suggest how these results might be representative (or not) of other areas in the world? For instance, would you expect to find the same results in countries where OSM is not as well mapped? I would also like to see discussion of using these maps for change analysis like ecosystem accounting. OSM has a significant weakness in that regard. I would also like to see more explanation of the spatial variation in accuracy scores seen in Figure 3.

The paper has many English language errors which should be corrected. I did not correct them in this review.

Detailed comments:

L2: In the title and throughout the manuscript I would argue that “LCLU” should be replaced with “LULC” (ie. Land-use/land-cover) to align with the majority use in the literature.

L10: replace “that: Can the” with “whether”

L11: Specify this was conducted in the UK.

L15: Here and throughout manuscript please avoid using generalized words like “high accuracy” without specifying a range or value. You should give the % because “high” can mean anything.

L24: “coast” is not referring to the water itself, but rather the shore and ocean. I would replace with “ocean”. I would also add reservoirs and canals to this list.

L26: The study by Niewenhuijsen that you cite here shows the exact opposite of your statement. I quote from their abstract “In this large prospective study (n = 792,649) in an urban area, we found … an increased risks of mortality with an increase in exposure to blue space (HR = 1.04, 95% CI 1.01⁻1.06 per 1%), NO₂ (HR = 1.01, 95% CI 1.00⁻1.02 per 5 ug/m³)”. Please revise this and be careful about generalized statements where you incorrectly cite literature.

L79: Remove question mark.

L99: How did you use GHS-SMOD to identify unique urban clusters/objects? How did you cluster pixels together? Please explain in more detail.

L101: Choosing a large number of cities does not minimize “subjectivity”. Rather it may reduce potential for bias. Please revise.

L111: What do you mean by positional accuracy? Is this for the OSM data? Please specify.

L116-144: As per my main comment above, it is problematic that you included flooded vegetation and wetland for some of the datasets and not for others. This needs to be addressed.

L151: Please elaborate on what tidal water means.

L177: If the buffer was 10m, then doesn’t this mean that the smallest water body width to be included is 11m and not 20m (assuming a pixel resolution of 1m)? Because the 11m water is buffered down to 1m with a -10m buffer and then buffered up again to 11m with a +10m buffer. If you buffer with 5m then the smallest water body with will be 6m? Please explain this process in more detail because as it stands it is confusing.

L199: This should be in the method section. And by do you use a 5m buffer as the reference instead of 0m? Please justify.

Figure 3: The color ramp for Accuracy and Precision needs to be tightened a bit because at the moment everything looks red and you cannot tell the difference between cities.

Figure 5: The box-and-whisker plots imply you are calculating accuracy at the city level and then aggregating up to the national level. Why don’t you provide a table showing the accuracy at the national level for each buffer instead? I think this is necessary because as it stands, the whiskers are so long that it is impossible to see the average accuracies in this figure.

Author Response

Response to Reviewer1

The authors present an evaluation of four global land-use land-cover maps for the mapping of urban blue space. They find that OSM performs better than the satellite remote sensing based maps, and is particularly good at mapping small (<10m width) water bodies. I find the results and conclusions very relevant and important, however I find a serious flaw in the analysis which needs to be addressed before the paper can be published. I explain below.

Reviewer1_Comment1
The main concern I have is that the authors use different definitions of “blue space” when selecting the LCLU classes form the global datasets. This results in confounded comparisons and incorrect accuracy assessments. Firstly, it is not clear why the authors include flooded vegetation and wetland for the Esri, ESA WorldCover and FROM-GLC10 maps, but they exclude wetland in the OSM map. This is inconsistent and needs to be addressed. I would argue that the authors come up with a more rigorous definition of “blue space” to include permanent water bodies that exclude wetlands (which are typically ephemerally covered with water). Secondly, if the authors are going to include flooded vegetation/wetland in their definition, then they need to also include mangroves for the ESA WorldCover definition because this aligns with the flooded vegetation definition in Esri.
Response to Reviewer1_Comment1: Thanks for this valuable comment! 
 First of all, the wetland in the OSM map has been included in the original manuscript.
 But in the revised manuscript, as suggested we excluded the types related to wetland (including 'flooded vegetation' and 'wetland') in various datasets. This is because wetland is seasonally flooded area that is a mix of grass/shrub/trees/bare ground, rather than permanent water bodies.
 Nevertheless, we found that the results (excluding wetland) reported in the revised manuscript are mostly consistency with those reported in the original manuscript. This is because the LULC type-wetland (or flooded vegetation) mostly has a small area percentage (see the below figure). 

Figure A. The area percentages of different LULC types.

Reviewer1_Comment2
I am also concerned that, as it stands, the discussion section is weak. The authors do not place their findings in the context of the other literature. They do not suggest how these results might be representative (or not) of other areas in the world? For instance, would you expect to find the same results in countries where OSM is not as well mapped? I would also like to see discussion of using these maps for change analysis like ecosystem accounting. OSM has a significant weakness in that regard. I would also like to see more explanation of the spatial variation in accuracy scores seen in Figure 3.
Response to Reviewer1_Comment2: Thanks for this valuable comment! In the revised manuscript, the 'discussion' part has been revised from the following aspects:

 First of all, we have discussed the limitation of using OSM dataset for blue space mapping. That is (see Section 5.2), 
 It remains necessary to validate our results with urban areas in other countries and regions. This is especially true for the OSM dataset, because existing studies have reported that both the map scale and the completeness (how well a region has been mapped) of OSM data may vary dramatically in different countries and regions (Touya and Reimer 2015; Tian et al. 2019; Wang et al. 2020; Zhou and Lin 2020; Zhou et al. 2022b). Specifically, Zhou et al. (2022b) reported that the completeness of OSM LULC data is higher in Europe than in other regions. This indicates that the findings when using OSM data may not be applicable to other study areas. Recall that this study involved only Great Britain as the study area due to the availability of the reference dataset. However, water bodies in rural areas and in other countries and regions should be considered for further validation.

 Second, we have discussed the use of these maps for change analysis. That is (see Section 5.1),
 Additionally, the ESRI data product includes data for each year from 2017–2021. Thus, this dataset may be used to analyze changes in land use (e.g., ecosystem accounting, Hein et al. 2015; Petersen et al. 2022). In contrast, both the ESA and FROM-GLC10 datasets are only available for a single year.

 Third, we have also compared our results with those in existing literatures (see Sections 5.1 and 5.2).
 Although other studies have compared different open LCLU datasets (Xu et al. 2019; Reinhart et al. 2021; Sun et al. 2022), most of these studies involved lower-resolution datasets (e.g., 20–100 m).
 These findings are not fully consistent with those of existing studies. For instance, Liao et al. (2021) reported that the FROM-GLC10 dataset gives the best performance for urban green space mapping in terms of accuracy, recall, and F1-score. This indicates that the effectiveness of using an LULC dataset may vary according to the application (e.g., blue space or green space mapping). 
 Mao et al. (2022) found that existing 30-m-resolution global water body datasets experience difficulties when mapping small rivers (i.e., widths narrower than 300 m).
 Specifically, Zhou et al. (2022b) reported that the completeness of OSM LULC data is higher in Europe than in other regions.

 Fourth, more explanation of the spatial variation in accuracy scores has been added for Figure 3. That is, 
 The accuracy is high (e.g., >90%) for most of the 133 urban areas, although this is not always the case for urban areas along the coastline.

Reviewer1_Comment3
The paper has many English language errors which should be corrected. I did not correct them in this review.
Response to Reviewer1_Comment3: Thanks for this valuable comment! We have used a language editing service (Charlesworth, https://www.cwauthors.com.cn/) to polish the language of this manuscript. 

Detailed comments:
Reviewer1_Comment4
L2: In the title and throughout the manuscript I would argue that “LCLU” should be replaced with “LULC” (ie. Land-use/land-cover) to align with the majority use in the literature.
Response to Reviewer1_Comment4: Thanks for this valuable comment! We have replaced the word “LCLU” with “LULC” across the manuscript.

Reviewer1_Comment5
L10: replace “that: Can the” with “whether”
Response to Reviewer1_Comment5: Thanks for this valuable comment! We have replaced the “that: Can the” with “whether”.

Reviewer1_Comment6
L11: Specify this was conducted in the UK.
Response to Reviewer1_Comment6: Thanks for this valuable comment! We have specified that this study was conducted in the UK.

Reviewer1_Comment7
L15: Here and throughout manuscript please avoid using generalized words like “high accuracy” without specifying a range or value. You should give the % because “high” can mean anything.
Response to Reviewer1_Comment7: Thanks for this valuable comment! We gave out the specific value (i.e., 98%) for “high accuracy”.

Reviewer1_Comment8
L24: “coast” is not referring to the water itself, but rather the shore and ocean. I would replace with “ocean”. I would also add reservoirs and canals to this list.
Response to Reviewer1_Comment8: Thanks for this valuable comment! 
 First, we replaced “coast” with “open sea”. 
 Second, we have also added “reservoirs” and “canals” in the revised manuscript.

Reviewer1_Comment9
L26: The study by Niewenhuijsen that you cite here shows the exact opposite of your statement. I quote from their abstract “In this large prospective study (n = 792,649) in an urban area, we found … an increased risks of mortality with an increase in exposure to blue space (HR = 1.04, 95% CI 1.01⁻1.06 per 1%), NO₂ (HR = 1.01, 95% CI 1.00⁻1.02 per 5 ug/m³)”. Please revise this and be careful about generalized statements where you incorrectly cite literature.
Response to Reviewer1_Comment9: Thanks for this valuable comment! There is a mistake here. Thus in the revised manuscript, 
 First, the corresponding literature has been removed. 
 Second, a relevant literature has been added. Specifically,

Reference
Zhu D, Zhou X. Effect of urban water bodies on distribution characteristics of particulate matters and NO2[J]. Sustainable Cities and Society, 2019, 50: 101679.

Reviewer1_Comment10
L79: Remove question mark.
Response to Reviewer1_Comment10: Thanks for this valuable comment! The question mark has been removed.

Reviewer1_Comment11
L99: How did you use GHS-SMOD to identify unique urban clusters/objects? How did you cluster pixels together? Please explain in more detail.
Response to Reviewer1_Comment11: Thanks for this comment! The GHS Settlement Model layers (GHS-SMOD) includes two datasets: the GHS-SMOD raster grid and the urban centre entities vector. Only the urban centre entities vector was acquired and used in our study. We have highlighted this point in the revised manuscript. That is, 
 These urban areas were freely acquired from the urban center vector dataset of the GHS Settlement Model (Florczyk et al. 2019).

Reviewer1_Comment12
L101: Choosing a large number of cities does not minimize “subjectivity”. Rather it may reduce potential for bias. Please revise.
Response to Reviewer1_Comment12: Thanks for this valuable comment! In the revised manuscript, the corresponding sentence has been revised. That is,
 Such a large number of urban areas, rather than only a few, were chosen to reduce bias in the analysis.

Reviewer1_Comment13
L111: What do you mean by positional accuracy? Is this for the OSM data? Please specify.
Response to Reviewer1_Comment13: Thanks for this comment! Yes, the ''positional accuracy'' was used for OSM data because OSM data is in a vector rather than a raster format.
We highlighted this point in the revised manuscript. That is,
 Although the OSM data may suffer from variable quality because they were edited by volunteers, previous studies have reported that the position accuracy of OSM data (in vector format) is mostly within 10 m (Haklay 2010; Zhou 2017; Brovelli and Zamboni 2018).

Reviewer1_Comment14
L116-144: As per my main comment above, it is problematic that you included flooded vegetation and wetland for some of the datasets and not for others. This needs to be addressed.
Response to Reviewer1_Comment14: Thanks for this valuable comment! 
 First of all, the wetland in the OSM map has been included in the original manuscript.
 But in the revised manuscript, as suggested we excluded the types related to wetland (including 'flooded vegetation' and 'wetland') in various datasets. This is because wetland is seasonally flooded area that is a mix of grass/shrub/trees/bare ground.
 Nevertheless, we found that the results (excluding wetland) are almost the same with those reported in the original manuscript. This is because the LULC type-wetland (or flooded vegetation) mostly has a small area percentage (see the below figure). 

Figure A. The area percentages of different LULC types.

Reviewer1_Comment15
L151: Please elaborate on what tidal water means.
Response to Reviewer1_Comment15: According to data product guide provided by Ordnance Survey (2019), TidalWater denotes the extends of tidal water, up to the High Water Mark and the Normal Tide Limit of rivers. We have given out the definition in the revised manuscript using a footnote. 

Reference
Ordnance Survey, (2019). OS Open Map - Local Product guide,https://www.ordnancesurvey.co.uk/business-government/tools-support/open-map-local-support, accessed May 2019.

Reviewer1_Comment16
L177: If the buffer was 10m, then doesn’t this mean that the smallest water body width to be included is 11m and not 20m (assuming a pixel resolution of 1m)? Because the 11m water is buffered down to 1m with a -10m buffer and then buffered up again to 11m with a +10m buffer. If you buffer with 5m then the smallest water body with will be 6m? Please explain this process in more detail because as it stands it is confusing.
Response to Reviewer1_Comment16: Thanks for this comment! If the buffer was 10m, the smallest water body width is 20m rather than 11m. This is because:
 First of all, the buffer method was only applied to the reference dataset (produced by the national mapping agency of Great Britain). Because this dataset is in a vector format, the width of the line symbol can be neglected. 
 Second, the buffer method was traditionally created on both sides of an object, https://pro.arcgis.com/en/pro-app/latest/tool-reference/analysis/buffer.htm. Thus the width is twice of the buffer radius. 

Reviewer1_Comment17
L199: This should be in the method section. And by do you use a 5m buffer as the reference instead of 0m? Please justify.
Response to Reviewer1_Comment17: Thanks for this comment! We used a 5m buffer radius (indicating a 10m width of water body) because:
 First of all, all the investigated raster datasets have a spatial resolution of 10m. 
 Second, the research objective (2) of this study is to investigate that whether a 10m-resolution LULC dataset can identify water bodies with a width of 10m. 

Reviewer1_Comment18
Figure 3: The color ramp for Accuracy and Precision needs to be tightened a bit because at the moment everything looks red and you cannot tell the difference between cities.
Response to Reviewer1_Comment18: Thanks for this comment! 
 First of all, the color looks red because the (classification) accuracy of water is relatively high (e.g., above 90%) for most urban areas.
 Nevertheless, the color ramp for Accuracy was set at 60-100% because the lowest accuracy was 69%; and that for Precision was set at 0-100% because the lowest precision was 6%.

Reviewer1_Comment19
Figure 5: The box-and-whisker plots imply you are calculating accuracy at the city level and then aggregating up to the national level. Why don’t you provide a table showing the accuracy at the national level for each buffer instead? I think this is necessary because as it stands, the whiskers are so long that it is impossible to see the average accuracies in this figure.
Response to Reviewer1_Comment19: Thanks for this valuable comment! A table corresponding to the Figure 5 has been given out. Please see Appendix A in the revised manuscript.

Reviewer 2 Report

This study evaluates and compares four LCLU datasets (ESRI, ESA, FROM-GLC10, OSM) for blue space (water bodies) mapping within urban areas. The topic and the developed methodology are of interest to some readers. However, the manuscript does not portray a high level of Originality/Novelty, nor significance of content.

Methodology:

Section 2.2 (Data) raises a major concern regarding the methodology for extracting LC types from the four datasets. The authors do not provide a convincing justification/criteria for considering LC types other than water as water, specifically, wetlands, riverbanks, and flooded vegetation. Besides, the manuscript does not evaluate the impact of this subjective decision on the outcome of the evaluation and comparison. The authors do mention this aspect in their conclusion (lines 404-405). This shortfall renders their findings of limited worth and practicality.

On a different matter, Section 2.2 (Data) provides some information regarding the four LULC datasets with citations to relevant literature. However, and for the sake of the general readership, it should have additionally provided information on the methodology used to produce these datasets and on the base/source for these datasets. Specifically, which satellite images and, more importantly, what are the resolutions for these images. A dataset claimed to be of 10m resolution does not mean necessarily that it was based on images of 10m or less resolution, hence, it may not be able to detect 10m objects. Consequently, assessing its capability to identify 10m objects becomes worthless.

On the same issue, the *scale* for the vector data OSM should have been mentioned and related to the resolution of the other raster datasets. Although a vector dataset does not have “resolution”, but it does not mean it is unlimited in terms of graphical representation of reality. Listing the source (and the scale) for the OSM dataset would have put it on a comparative terms with the other three datasets.

Typographical Errors:

Lines 192-194: It seems that the FN and FP were used the wrong way round. To my understanding, FP should be used to denote the total area of water bodies that in the open dataset but not in the reference dataset, while FN should be used to denote the total area of water bodies that in the reference dataset but not in the open dataset. Only by this way, the 3rd conclusion (lines 214-227) regarding the recall would be valid.

Line 166: correct the caption in for Figure 2 (b) and (c); 1th -> 1st and 2th -> 2nd

Line 222: “the three LC datasets” is ambiguous; specify which three LC datasets

Line 293: “the three LC datasets” is ambiguous; specify which three LC datasets

Line 397: “the three LC datasets” is ambiguous; specify which three LC datasets especially that OSM and FROM-GLC10 are the other two datasets mentioned in this paragraph.

Lines 190-194: Write the full definition for TP, TN, FN, and FP prior to using the abbreviations.

Author Response

Response to Reviewer2

Reviewer2_Comment1

This study evaluates and compares four LCLU datasets (ESRI, ESA, FROM-GLC10, OSM) for blue space (water bodies) mapping within urban areas. The topic and the developed methodology are of interest to some readers. However, the manuscript does not portray a high level of Originality/Novelty, nor significance of content.

Response to Reviewer2_Comment1: Thanks for this valuable comments! This study has several contributions.

  • In terms of methodology, a buffer approach was proposed to extract water bodies of different widths from a reference dataset.
  • In terms of results, this study
  • not only evaluates and compares a total of four LULC datasets (ESRI, ESA, FROM-GLC10, OSM) for blue space mapping,
  • but also investigates that whether a 10m-resolution LULC dataset can identify water bodies with a width of 10m.

Reviewer2_Comment2

Methodology:

Section 2.2 (Data) raises a major concern regarding the methodology for extracting LC types from the four datasets. The authors do not provide a convincing justification/criteria for considering LC types other than water as water, specifically, wetlands, riverbanks, and flooded vegetation. Besides, the manuscript does not evaluate the impact of this subjective decision on the outcome of the evaluation and comparison. The authors do mention this aspect in their conclusion (lines 404-405). This shortfall renders their findings of limited worth and practicality.

Response to Reviewer2_Comment2: Thanks for this valuable comment! In the revised manuscript,

  • First of all, as suggested by another reviewer, we excluded the types related to wetland (including 'flooded vegetation' and 'wetland') in various datasets. This is because wetland is seasonally flooded area that is a mix of grass/shrub/trees/bare ground, rather than permanent water bodies.
  • This point has been discussed in the revised manuscript. That is,
  • there are a number of LULC types in each LULC dataset. We mainly extracted areas of water from the LC datasets, and used the five different types of water bodies (dock, reservoir, river, riverbank, and water) in the OSM dataset. These types are conceptually defined as permanent water areas, rather than seasonal water areas (e.g., wetland)..

  • Second, we found that the results (excluding wetland) reported in the revised manuscript are mostly consistency with those reported in the original manuscript. This is because the LULC type-wetland (or flooded vegetation) mostly has a small area percentage (see the below figure).

Figure A. The area percentages of different LULC types.

Reviewer2_Comment3

On a different matter, Section 2.2 (Data) provides some information regarding the four LULC datasets with citations to relevant literature. However, and for the sake of the general readership, it should have additionally provided information on the methodology used to produce these datasets and on the base/source for these datasets. Specifically, which satellite images and, more importantly, what are the resolutions for these images. A dataset claimed to be of 10m resolution does not mean necessarily that it was based on images of 10m or less resolution, hence, it may not be able to detect 10m objects. Consequently, assessing its capability to identify 10m objects becomes worthless.

Response to Reviewer2_Comment3: Thanks for this valuable comment! All the three raster datasets were produced using a 10m resolution remote sensing data (Sentinel-2) as the basis. We have highlighted this point in the revised manuscript. That is,

  • Specifically, the three raster datasets (ESRI, ESA, FROM-GLC10) were all produced using 10-m-resolution RS data (Sentinel-2) as the source.

Reviewer2_Comment4

On the same issue, the *scale* for the vector data OSM should have been mentioned and related to the resolution of the other raster datasets. Although a vector dataset does not have “resolution”, but it does not mean it is unlimited in terms of graphical representation of reality. Listing the source (and the scale) for the OSM dataset would have put it on a comparative terms with the other three datasets.

Response to Reviewer2_Comment4: Thanks for this comment! The map scale of OSM data may vary not only in different regions but also for different objects. This is because the (OSM) data were edited by volunteers across the globe and they may suffer from the quality issue.

  • Thus in the revised manuscript, we have discussed the limitation of using OSM data.
  • It remains necessary to validate our results with urban areas in other countries and regions. This is especially true for the OSM dataset, because existing studies have reported that both the map scale and the completeness (how well a region has been mapped) of OSM data may vary dramatically in different countries and regions (Touya and Reimer 2015; Tian et al. 2019; Wang et al. 2020; Zhou and Lin 2020; Zhou et al. 2022b). Specifically, Zhou et al. (2022b) reported that the completeness of OSM LULC data is higher in Europe than in other regions. This indicates that the findings when using OSM data may not be applicable to other study areas. Recall that this study involved only Great Britain as the study area due to the availability of the reference dataset. However, water bodies in rural areas and in other countries and regions should be considered for further validation.

Reference

Touya, G., and Reimer, A., (2015). Inferring the scale of OpenStreetMap features. In OpenStreetMap in GIScience, edited by Arsanjani, J. J., Zipf, A., Mooney, P., and Helbich, M., 81-99. Lecture Notes in Geoinformation and Cartography.

  • Despite of the (OSM) quality issue, we have also highlighted in the revised manuscript that existing studies have reported the OSM data have a relatively high positional accuracy (smaller than 10m). That is,
  • Although the OSM data may suffer from variable quality because they were edited by volunteers, previous studies have reported that the position accuracy of OSM data (in vector format) is mostly within 10 m (Haklay 2010; Zhou 2017; Brovelli and Zamboni 2018).

Typographical Errors:

Reviewer2_Comment5

Lines 192-194: It seems that the FN and FP were used the wrong way round. To my understanding, FP should be used to denote the total area of water bodies that in the open dataset but not in the reference dataset, while FN should be used to denote the total area of water bodies that in the reference dataset but not in the open dataset. Only by this way, the 3rd conclusion (lines 214-227) regarding the recall would be valid.

Response to Reviewer2_Comment5: Thanks for this valuable comment! The mistake has been revised in the manuscript. That is,

  • TP (true positive) denotes the common area between water bodies extracted from an open LULC dataset and the reference dataset, TN (true negative) denotes the common area between non-water bodies (regions not classified as water bodies) in both datasets, FP (false positive) denotes the total area of water bodies extracted from the open dataset but not from the reference dataset, and FN (false negative) denotes the total area of water bodies extracted from the reference dataset but not from the open dataset.

Reviewer2_Comment6

Line 166: correct the caption in for Figure 2 (b) and (c); 1th -> 1st and 2th -> 2nd

Line 222: “the three LC datasets” is ambiguous; specify which three LC datasets

Line 293: “the three LC datasets” is ambiguous; specify which three LC datasets

Line 397: “the three LC datasets” is ambiguous; specify which three LC datasets especially that OSM and FROM-GLC10 are the other two datasets mentioned in this paragraph.

Lines 190-194: Write the full definition for TP, TN, FN, and FP prior to using the abbreviations.

Response to Reviewer2_Comment6: Thanks for this valuable comment! In the revised manuscript,

  • First, the caption in Figure 2 have been revised.
  • Second, “the three LC datasets” have been specified in Line 222, Line 293 and Line 397.
  • Third, the full definitions for TP, TN, FN, and FP have been given out.

Reviewer 3 Report

This paper presents a very interesting and useful topic.

It is well-written and balanced. The findings, in particular, Fig. 5 and 6 and their descriptions provide very useful information for potential data users.

I have only few recommendatios for the authors:

Lines 89-93 In my opinion, this paragraphs is not necessary since the paper has a standard scientific article structure.

Lines 147-152 Please provide more details on the reference data (raster/vector, spatial resolution, classification...in order to better understand the results

Discussion is rather short and I suggest to provide more existing use cases of blue spaces mapping with pros and cons of the analysed datasets with relevant references.  What is the value of those datasets for change mapping (of blue spaces). At what scale(s) would the authors recommend to use the datasets - local, regional, or higher? For urban planners or manager they might be too coarse, for landscape/environmental analyses they might have issues in change detection. Please comment in Discussion or Conclusion.

Author Response

Response to Reviewer3

This paper presents a very interesting and useful topic.
It is well-written and balanced. The findings, in particular, Fig. 5 and 6 and their descriptions provide very useful information for potential data users.
I have only few recommendatios for the authors:

Reviewer3_Comment1
Lines 89-93 In my opinion, this paragraphs is not necessary since the paper has a standard scientific article structure.
Response to Reviewer3_Comment1: Thanks for this valuable comment! We have removed this paragraph as suggested.

Reviewer3_Comment2
Lines 147-152 Please provide more details on the reference data (raster/vector, spatial resolution, classification...in order to better understand the results
Response to Reviewer3_Comment2: Thanks for this valuable comment! In the revised manuscript, more details on the reference data have been given out. Specifically,
 These data were acquired in vector format at a 1:10,000 scale; more importantly, they are “the most detailed ‘street level’ mapping product available within the open data arena” (Ordnance Survey 2019).

Reviewer3_Comment3
Discussion is rather short and I suggest to provide more existing use cases of blue spaces mapping with pros and cons of the analysed datasets with relevant references. What is the value of those datasets for change mapping (of blue spaces). At what scale(s) would the authors recommend to use the datasets - local, regional, or higher? For urban planners or manager they might be too coarse, for landscape/environmental analyses they might have issues in change detection. Please comment in Discussion or Conclusion.
Response to Reviewer3_Comment3: Thanks for this valuable comment! In the revised manuscript, the 'discussion' part has been revised from the following aspects:

 First of all, we have discussed the limitation of using OSM dataset for blue space mapping. That is (see Section 5.2), 
 It remains necessary to validate our results with urban areas in other countries and regions. This is especially true for the OSM dataset, because existing studies have reported that both the map scale and the completeness (how well a region has been mapped) of OSM data may vary dramatically in different countries and regions (Touya and Reimer 2015; Tian et al. 2019; Wang et al. 2020; Zhou and Lin 2020; Zhou et al. 2022b). Specifically, Zhou et al. (2022b) reported that the completeness of OSM LULC data is higher in Europe than in other regions. This indicates that the findings when using OSM data may not be applicable to other study areas. Recall that this study involved only Great Britain as the study area due to the availability of the reference dataset. However, water bodies in rural areas and in other countries and regions should be considered for further validation.

 Second, we have discussed the use of these maps for change analysis. That is (see Section 5.1),
 Additionally, the ESRI data product includes data for each year from 2017–2021. Thus, this dataset may be used to analyze changes in land use (e.g., ecosystem accounting, Hein et al. 2015; Petersen et al. 2022). In contrast, both the ESA and FROM-GLC10 datasets are only available for a single year.
 a recent study reported that the ESRI dataset can be used to acquire urban blue space for investigating land surface temperature (Chen et al. 2022).

 Third, we have also compared our results with those in existing literatures (see Sections 5.1 and 5.2).
 Although other studies have compared different open LCLU datasets (Xu et al. 2019; Reinhart et al. 2021; Sun et al. 2022), most of these studies involved lower-resolution datasets (e.g., 20–100 m).
 These findings are not fully consistent with those of existing studies. For instance, Liao et al. (2021) reported that the FROM-GLC10 dataset gives the best performance for urban green space mapping in terms of accuracy, recall, and F1-score. This indicates that the effectiveness of using an LULC dataset may vary according to the application (e.g., blue space or green space mapping). 
 Mao et al. (2022) found that existing 30-m-resolution global water body datasets experience difficulties when mapping small rivers (i.e., widths narrower than 300 m).
 Specifically, Zhou et al. (2022b) reported that the completeness of OSM LULC data is higher in Europe than in other regions.

Round 2

Reviewer 1 Report

The authors have adequately addressed my critiques and the manuscript is now ready for publication. It will be a valuable contribution to the field.